# Bayesian Group Index Regression for Modeling Chemical Mixtures and Cancer Risk

**DOI:** 10.3390/ijerph18073486

**Published:** 2021-03-27

**Authors:** David C. Wheeler, Salem Rustom, Matthew Carli, Todd P. Whitehead, Mary H. Ward, Catherine Metayer

**Affiliations:** 1Department of Biostatistics, School of Medicine, Virginia Commonwealth University, Richmond, VA 23298-0032, USA; rustoms@mymail.vcu.edu (S.R.); carlimm@mymail.vcu.edu (M.C.); 2UC Berkeley School of Public Health, University of California, Berkeley, CA 94704-7394, USA; ToddPWhitehead@Berkeley.edu (T.P.W.); cmetayer@berkeley.edu (C.M.); 3Occupational and Environmental Epidemiology Branch, Division of Cancer Epidemiology and Genetics, National Cancer Institute, Rockville, MD 20850, USA; wardm@exchange.nih.gov

**Keywords:** mixture analysis, environment, cancer, chemical mixtures

## Abstract

There has been a growing interest in the literature on multiple environmental risk factors for diseases and an increasing emphasis on assessing multiple environmental exposures simultaneously in epidemiologic studies of cancer. One method used to analyze exposure to multiple chemical exposures is weighted quantile sum (WQS) regression. While WQS regression has been demonstrated to have good sensitivity and specificity when identifying important exposures, it has limitations including a two-step model fitting process that decreases power and model stability and a requirement that all exposures in the weighted index have associations in the same direction with the outcome, which is not realistic when chemicals in different classes have different directions and magnitude of association with a health outcome. Grouped WQS (GWQS) was proposed to allow for multiple groups of chemicals in the model where different magnitude and direction of associations are possible for each group. However, GWQS shares the limitation of WQS of a two-step estimation process and splitting of data into training and validation sets. In this paper, we propose a Bayesian group index model to avoid the estimation limitation of GWQS while having multiple exposure indices in the model. To evaluate the performance of the Bayesian group index model, we conducted a simulation study with several different exposure scenarios. We also applied the Bayesian group index method to analyze childhood leukemia risk in the California Childhood Leukemia Study (CCLS). The results showed that the Bayesian group index model had slightly better power for exposure effects and specificity and sensitivity in identifying important chemical exposure components compared with the existing frequentist method, particularly for small sample sizes. In the application to the CCLS, we found a significant negative association for insecticides, with the most important chemical being carbaryl. In addition, for children who were born and raised in the home where dust samples were taken, there was a significant positive association for herbicides with dacthal being the most important exposure. In conclusion, our approach of the Bayesian group index model appears able to make a substantial contribution to the field of environmental epidemiology.

## 1. Introduction

There are more than 80,000 chemicals on the market in the United States alone and some are found in many consumer products [1]. Hence, individuals are exposed to chemical mixtures daily. Traditionally, epidemiologic studies of cancer and environmental chemical exposures have evaluated chemicals independently using a single-chemical regression approach [2,3,4,5,6,7,8]. More recently, there has been a growing interest in the literature on understanding the joint effects of multiple environmental risk factors for diseases [9,10,11,12] and an increasing emphasis on assessing multiple environmental exposures simultaneously in epidemiologic studies of cancer [13,14]. In this paper, we focus on exposure to multiple diverse environmental chemicals and develop a statistical method that expands on weighted quantile sum regression (WQS) [14] to model cancer risk. WQS regression was developed to identify the truly “bad actors” when modeling exposure to a chemical mixture in a risk assessment setting. This constrained regression method is designed to accommodate highly correlated data that create collinearity issues with traditional regression methods. In WQS regression, a weighted index of exposures is estimated, where the weights for each chemical exposure are constrained to be between 0 and 1 and sum to 1. This approach has been used in many studies of environmental mixtures and health outcomes. For example, WQS was used to model non-Hodgkin lymphoma risk related to a mixture of 27 chemicals in the NCI-SEER NHL study [15].

While WQS regression has been demonstrated to have good sensitivity and specificity when identifying important exposures [14,16], it has certain limitations. One limitation of WQS regression is that it uses a two-step model fitting process and a splitting of data into training and validation sets that decreases power and stability with small datasets that are common in epidemiology. Another limitation of WQS regression is that all chemical exposures in the weighted index are constrained to have associations in the same direction with the outcome. This constraint does not allow for the realistic situation when chemicals in different classes have different associations with a health outcome in both direction and magnitude. For example, there is evidence that insecticides have a negative association with non-Hodgkin lymphoma (NHL) [2], while organochlorine compounds such as some PCB congeners have a positive association with NHL [3]. Considering the multitude of diverse chemicals to which individuals are exposed daily, more flexible approaches to modeling environmental cancer risk are needed.

To overcome the single-index limitation of WQS regression, we have proposed grouped weighted quantile sum (GWQS) regression to enable multiple groups of chemicals in the model, where each chemical group can have a different magnitude and direction of association with the outcome [17,18]. GWQS moves the analytical approach to environmental risk assessment toward more realistic models of environmental exposures by estimating a weighted index for each group of exposures. A simulation study of GWQS demonstrated that it had better power, sensitivity, specificity, and goodness-of-fit than WQS when there were two or more groups of exposures [19]. GWQS also performed better overall than lasso and the group lasso with a minimax concave penalty. This simulation study showed the inability of both lasso and WQS regression to estimate exposure effects for realistic mixtures with different groups of chemicals that were positively and negatively associated with risk. Both WQS and lasso produced an effect estimate that averaged over positive and negative effects, resulting in effect estimates that were biased toward the null. While this assessment was encouraging for the application of GWQS in studies of environmental cancer risk, GWQS still has the limitation of a two-step estimation process and splitting of data into training and validation sets, which can result in reduced power in small epidemiologic studies.

We propose to use a Bayesian framework to create a more flexible and complex GWQS model that does not require two-step estimation. We have previously used Bayesian index regression to create single-index models of neighborhood deprivation and risk of elevated blood lead levels [20,21] and tobacco retail outlet rates [22]. In this paper, we extend the Bayesian index model to incorporate multiple exposure indices (similar to GWQS regression) and term the approach the Bayesian group index model, a new way to estimate the health effects of chemical mixtures.

## 2. Materials & Methods

### 2.1. Bayesian Group Index Regression

The basic Bayesian index regression model for a binary health outcome $y_{i}\sim \mathrm{Bernoulli}(p_{i})$ is specified through the log-odds of disease for the *i*th subject as
(1)$$\mathrm{logit}(p_{i})=\beta_{0}+\beta_{1}\left(\sum_{j=1}^{C}w_{j}q_{ij}\right)+z_{i}^{T}\phi$$
where the left-hand side of the equation is the logit of the disease probability $p_{i}$, $w_{j}$ is the weight parameter for the jth exposure with quantile score $q_{ij}$ for the *i*th individual, *β*_1_ is the effect for the index, and $z_{i}^{T}$ is a vector of covariates with corresponding effects in vector ϕ. Quantiles are used instead of raw data to reduce the effect of outliers and account for different concentration scaling for different exposures. Any reasonable definition of quantiles could be used, including deciles. In this regression model there is one weighted index using C number of exposures. The weights $w_{j}$ represent relative importance of the exposures and are constrained to be between 0 and 1 and to sum to 1. Assignment of distributions for the model parameters completes the model specification. The index weights $w_{1},\ldots ,w_{C}$ are given a Dirichlet prior with parameters **α** = (*α*_1_, …, *α_c_*). The Dirichlet prior is convenient because it assures that the weights $w_{j}\in (0,1)$ and $\sum_{j=1}^{C}w_{j}=1$. The intercept, index regression coefficient, and covariate regression coefficients are assigned vague normal priors, $\beta_{1}\sim Normal\left(0,\;\tau_{1}\right)$ with precision $\tau_{1}=1/\sigma_{1}^{2}$ and $\sigma_{1}\sim Uniform(0,100)$. An improper uniform distribution α~dflat() could also be used for the intercept, particularly if random effects are included in the model.

To better model multiple sets of diverse environmental exposures, we extend the Bayesian index model to a Bayesian group index model that allows for multiple exposure groups, each with potentially different direction and magnitude of association with the health outcome. The Bayesian group index model includes a weighted exposure index and associated effect for each exposure group. For example, a model for three groups of exposures is
(2)$$\mathrm{logit}(p_{i})=\beta_{0}+\beta_{1}\left(\sum_{j=1}^{C_{1}}w_{j1}q_{ij1}\right)+\beta_{2}\left(\sum_{j=1}^{C_{2}}w_{j2}q_{ij2}\right)+\beta_{3}\left(\sum_{j=1}^{C_{3}}w_{j3}q_{ij3}\right)+z_{i}^{T}\phi$$
where $w_{j1}$ is the weight for the *j*th exposure in the first index, $q_{ij1}$ is the quantile for the *j*th exposure in the first index for the *i*th subject, and the weights and quantiles are defined similarly for the second and third exposure indexes. There is a variable number $C_{k}$ of exposures in each index and each index has a regression coefficient $\beta_{k}$ (*k* = 1, 2, 3 in this example). This model can identify the most important among the groups of exposures through posterior inference on the index effects $\beta_{1},\beta_{2},\beta_{3}$ and the most important variables in each index through posterior inference on the weights.

The priors for the parameter in this model are similar to the base Bayesian index regression model, with the single index priors extended for multiple groups. The weights for each index follow a Dirichlet prior and the index effect for each group follows a vague normal prior. Different choices of priors are possible for the index effect parameters. For example, a mixture prior with a penalty could be used for the index effects to overcome any model instability due to collinearity [23]. This model could also be extended to include a subject-level random effect $\psi_{i}$ to account for residual confounding at the individual level, and a natural choice for the prior would be $\psi_{i}\sim Normal(0,\tau_{\psi })$ with precision $\tau_{\psi }=1/\sigma_{\psi }^{2}$ and $\sigma_{\psi }\sim Uniform(0,100)$. Markov chain Monte Carlo (MCMC) is used to estimate the model parameters. Convergence of the MCMC algorithm is done using the Gelman-Rubin or Geweke diagnostic statistics. Our implementation of the Bayesian group index regression model is available in an R package titled BayesGWQS [24] to facilitate use by other researchers.

### 2.2. Simulation Study Design

To assess the performance of the Bayesian group index model, we generated chemical concentration data over several different exposure scenarios, which varied in the amount of chemical correlation, the number of chemical groups, and the strength of association between each group and the outcome. There were four scenario sets (A–D) that varied in total group number and number of chemicals per group. We also considered different exposure effect strengths (Strengths 1–5). For each scenario set, we started with a null effect of odds ratio (OR) = 1.00 and then increased the association in strength for each chemical group (both negative and positive associations). In scenario sets A–C, for positive associations the strengths 2–5 denote ORs of 1.50, 2.00, 2.50, and 3.00, respectively, while negative associations were the reciprocals of these ORs (0.67, 0.50, 0.40, 0.33, respectively). To evaluate the ability of the models to estimate smaller true effects, the positive (negative) scenario set D had effect sizes of 1.00 (1.00), 1.25 (0.80), 1.50 (0.67), 1.75 (0.57), and 2.00 (0.50), respectively. Moreover, the sample size in scenario set D was reduced to 500 from 1000 in scenario sets A-C to evaluate model performance in smaller studies such as the California Childhood Leukemia Study (CCLS).

For the correlation amongst the chemicals, a weak, moderate, and strong correlation structure was considered in each scenario set. These three correlation structures were: weak (W) with correlation of 0.5 within group and 0.1 across group, moderate (M) with correlation of 0.7 within group and 0.3 across group, and strong (S) with correlation of 0.9 within group and 0.5 across group. Correlation structures for the chemicals were specified through a covariance matrix. This covariance matrix was generated via a vector of means and a vector of standard deviations that also allowed for generation of the data as multivariate normal. Four quantiles of the exposures were used in all simulations for computation of the weighted index in each group (e.g., $q_{ij}=0,1,2,3$).

For the first scenario set with 9 chemicals (Scenario set A), we generated five different chemicals in one group and four different chemicals in another group. The first group had a negative association with the outcome and the second group had a positive association (except for the null effect scenarios). Through setting of the chemical weights, two chemicals in each group were specified to be important and the remaining were set to be not important. The important chemicals in each group were given equal weight, with the weights summing to 1 for each group (e.g., two important chemicals in a group would lead to each having a weight of 0.5). Unimportant chemicals were assigned a true weight of 0.

For the second scenario set (Scenario set B), there were 14 chemicals allocated among three groups. Group 1 had a negative association and groups 2 and 3 each had a positive association with the outcome (excluding the null effects scenario). There was one important chemical in each group. The third scenario set (Scenario set C) was similar to Scenario set B except that group 2 had two important chemicals and groups 1 and 3 each had three important chemicals. Scenario set D had the same group structure as scenario set C. The different terms used in the simulation scenarios are summarized in Table 1. The terms are used to succinctly present the simulation study results. Each individual scenario is defined in Appendix A.

Based on the defined exposure scenarios, we replicated a case–control study with a relatively balanced number of cases and controls (50 ± 10% cases) for a binary outcome *y* in every iteration of the data generation. The outcome was distributed as y~Binomial(n,p) where $p=\frac{1}{1+e^{\eta }}$ and $\eta =\beta_{0}^{*}+\sum_{j=1}^{K}\beta_{j}^{*}[WQS_{j}^{*}]$ and the star notation indicates the true parameter values. As no covariates were used to generate the data, the term $z^{T}\phi =0$. We simulated 100 data sets for each scenario to replicate 100 studies.

To assess the performance of the Bayesian group index model in comparison to GWQS, we calculated the power, bias, and mean squared error (MSE) of the exposure effects for each of the groups, as well as the specificity and sensitivity for identifying unimportant versus important chemicals. When calculating power, we examined the proportion of 95% credible (or confidence for GWQS) intervals of the odds ratios of chemical group associations that did not contain 1.00. We calculated sensitivity as the proportion of truly important chemicals that the model identified as important. This was done by identifying if the weights of the important chemicals were estimated to be greater than or equal to $\frac{1}{C_{j}}$. Specificity was defined as the proportion of the truly unimportant chemicals that were correctly identified as unimportant by the models. We defined a chemical as unimportant if its weight was estimated to be less than the threshold of $\frac{1}{C_{j}}$. We fitted the Bayesian group index models using our R package BayesGWQS [24] and fitted the GWQS models using our R package groupWQS [18]. A vignette for groupWQS is available on The Comprehensive R Archive Network [18] that demonstrates use of the package.

### 2.3. Data Analysis

To apply the Bayesian group index model to observed data, we analyzed childhood leukemia risk in the CCLS, which is a population-based case–control study conducted in 18 counties in the Central Valley and 17 counties within the San Francisco Bay area and designed to assess the relationships between genetic factors and environmental exposures and childhood leukemia [25]. Cases were identified within three days after diagnosis from 1995 to 2012 from nine pediatric clinical centers in the study area. Inclusion criteria included: (1) residence in California at the time of diagnosis, (2) without prior cancer diagnosis, (3) age under 15 years, and (4) having a Spanish- or English-speaking biological parent. Controls were selected from state birth certificate files and matched to cases on sex, date of birth, Hispanic ethnicity, and (maternal) race.

Participating parents were initially interviewed to ascertain information about their children’s exposure to potential risk factors for leukemia. A subset of the families participated in a second in-home interview during which dust samples were collected. Dust was collected from homes of controls and cases who were younger than 8 years at the time of diagnosis (similar reference date for controls) who were living at the diagnosis home. The condition of living in the diagnosis home was used so that the dust sample from carpet would represent the exposures over a significant part of the early life of a child. A total of n = 583 children participated after the second interview, of which 277 were cases and 306 were controls.

Dust samples were collected from a rug or carpet in the room where the child spent the most time while awake (commonly the family room) by a high-volume small surface sampler (HVS3) and/or from the household vacuum cleaner. Colt et al. [26] found the household vacuum to be a valid alternative to the HVS3 for detecting, ranking, and quantifying concentrations of pesticides and other compounds. After extraction, concentrations of 64 organic chemicals were measured using gas chromatography/mass spectrometry [26]. Nine metals were measured using inductively coupled plasma/mass spectrometry (ICP/MS) combined with microwave-assisted acid digestion. After excluding participants due to missing covariate information, 296 controls and 268 cases were included in this analysis (n = 564). We used exposures for 49 chemicals (Appendix A) where at least one-fifth of the measurements were above the detectable limit. The chemical concentrations that were below this limit were imputed between 0 and the limit of detection using univariate imputation with the assumption of a lognormal distribution.

The concentrations for some of the pairs of chemicals measured in dust were observed to be strongly correlated. The chemicals that had the strongest correlation with each other were found to be in the same class of chemical. As an example, several of the PAHs were highly correlated (e.g., r = 0.90 for chrysene and benzo[a]anthracene). In addition, congeners or chemicals within the following classes of chemicals were highly correlated: PCBs, organochlorine insecticides, and pyrethroid insecticides. Such strong correlations observed between chemicals in these classes makes modeling simultaneous chemical exposure effects untenable via traditional regression methods. In this case, the use of mixture analysis methods such as the Bayesian group index model is warranted.

To analyze the association between chemical exposure and childhood leukemia, we put the 49 chemicals into the following groups: insecticides, PAHs, PCBs, metals, herbicides, and the tobacco exposure markers of cotinine and nicotine. These groups were based on their use (e.g., insecticides, herbicides) or structural similarity (PCBs, PAH). The fungicide ortho-phenylphenol was placed in the herbicide group. We then estimated the risk of childhood leukemia associated with each of the six exposure groups simultaneously using the Bayesian group index model while adjusting for the following covariates: child’s sex, age, ethnicity, annual household income, mother’s age at birth of child, mother’s education level, and if the child had lived in the dust sampling residence since the time of birth. In addition, we performed a stratified analysis using the binary variable for whether the child had lived in the dust sample home since birth as the stratifying variable instead of an adjustment variable. This was done to determine if the exposure effects were greater for those whose who lived in the same house (where the dust samples had been taken) since birth. There were n = 279 children who lived in the same house since birth and n = 285 who did not. In fitting the models, we used quartiles of exposures and 15,000 Markov chain Monte Carlo (MCMC) iterations with two chains and 5000 iterations as a burn-in sample. Convergence of all parameters in the model was verified via the Gelman-Rubin diagnostic statistic (i.e., upper CI was less than 1.10). We summarized the results through posterior mean estimates of the ORs and 95% credible intervals for each exposure group and also with forest plots. We identified the important chemical exposures in each chemical group using the posterior mean weight estimates and visualized them via weight plots. Model fitting was done using our R package BayesGWQS [24]. Study protocols involving research with human participants were approved by the institutional review boards at the University of California, Berkeley, the National Cancer Institute, and Virginia Commonwealth University.

## 3. Results

### 3.1. Simulation Study

We present results for scenarios set D in the main paper, while scenario set A–C results are in the Appendix A. The estimated odds ratios and power for the Bayesian group index model and GWQS regression in scenario set D are listed in Table 2. Both the Bayesian group index and GWQS models accurately estimated the odds ratio across different strengths of association and chemical correlations. In scenario sets A-C, GWQS was slightly closer to the true odds ratio (Appendix A), particularly when there was weak or moderate chemical correlation and strong association (true OR = 3.0). In these scenarios, the Bayesian model tended to estimate higher odds ratios for the positive association chemical groups than did GWQS. Type I error rates were similar, but slightly higher for GWQS. The most notable different between models was in power, where the Bayesian group index model was consistently more powerful than the GWQS model in scenario set D. For example, with moderate correlation structure and OR of 1.50 or 0.67, the power for each of the chemical group coefficients is around 0.91 to 0.95 for the Bayesian group index model while it is between 0.62 and 0.69 for the GWQS regression model. However, when the sample size and effect sizes were larger in scenario sets A-C, differences were smaller and both methods reached a power of (or near) 1.00 with larger true strengths of association (Appendix A). As expected, true stronger associations led to higher power in both models. These findings were consistent with the results for scenario set A and B in Appendix A, respectively. The models in scenario set C and D had greater power than scenario set B due to increased signal from having multiple rather than a single important chemical(s).

Table 3 compares the bias and MSE of both models for scenario set D. Appendix A contain the bias and MSE for scenario sets A–C. Most of the associations with true OR > 1 had positive bias in the Bayesian group index model in scenario sets A–C, but this did not occur as often in scenario set D. While the bias appears to be slightly larger in the Bayesian group index model, the MSE is also smaller. The slightly larger bias found in the Bayesian index model reflects the estimated odds ratio findings. As expected, in both models a larger true odds ratio generally led to larger MSE values across scenarios.

Table 4 compares the sensitivity and specificity found in GWQS and the Bayesian group index models for scenario set D, while Appendix A displays these for the remaining scenario sets (A–C). We see similar patterns in both models when the correlation structure gets stronger. Outside of the null effect, as the correlation structure gets stronger both the sensitivity and specificity for Bayesian group index model and GWQS decrease. This is because stronger correlation in the predictors makes identification of the important chemicals more difficult. As expected, a larger odds ratio led to larger values in sensitivity and specificity in both models. Outside of the null effect and small effect sizes, sensitivity and specificity were better with the Bayesian group index model overall. When there were multiple important chemicals in each group (scenario sets C and D) instead of one important chemical per group (scenario set B), the specificity increased. When going from two groups (scenario set A) to three groups (scenario set B) in the mixture, the specificity decreased. This suggests that the model performance decreases with increasing number of groups in the mixture, but that it increases as the number of important members in each group increases.

### 3.2. Application to Childhood Leukemia Risk Estimates

A summary of the demographics for children with and without leukemia from the CCLS study is presented in Table 5. Child’s age, sex, and mother’s age were equally distributed between cases and controls, while more control children were in the highest household income bracket (53.7%) compared to cases (39.9%). In addition, mothers of cases had slightly lower rates of post-secondary education (39.2%) compared to controls (45.6%). A larger proportion of controls resided in the same residence since birth (53.7%), compared to cases (44.8%).

The odds ratios for childhood leukemia associated with each of the six exposure groups calculated from the Bayesian group index model for the CCLS are in Table 6. Insecticides had a significant negative effect indicated by an odds ratio of 0.64 (95% CI: 0.40, 0.99). PCBs, PAHs, and herbicides had positive effects that were not significant according to the 95% credible intervals. Metals and tobacco markers had inverse but not statistically significant effects. The pattern in effects is clearly visible in the forest plot of the estimates in Figure 1. The variability was greatest for herbicides according to the credible intervals. The estimated weights of the chemical components for each group are plotted in Figure 2. Among insecticides, carbaryl was overwhelmingly the most important chemical with a posterior mean weight of 0.144. The highest category of household income (USD 75,000 or more) was associated (OR = 0.36) with significantly reduced leukemia risk, while living in the sampling household since birth (OR = 0.69) was associated with lowered likelihood of childhood leukemia. Age (child and mother’s), sex, ethnicity, and mother’s education were not significantly associated with childhood leukemia incidence.

Results of the stratified analysis with only children with a different residence since birth show that no chemical groups were found to have a significant association with childhood leukemia, although household income was still inversely associated with childhood leukemia risk (Appendix A). However, for children that had the same residence since birth (Table 7), herbicides had a significant positive association with childhood leukemia (OR = 2.22, 95% CI: 1.45, 3.61). In addition, insecticides were found to have a stronger (yet more variable) negative association with childhood leukemia (OR = 0.50, 95% CI: 0.23, 0.99) than with the non-stratified analysis. Forest plots visualize the chemical group associations for childhood leukemia for children who changed residence since birth in Appendix A and for children with the same residence since birth in Figure 3. The estimated weights of the chemical components for the stratified analyses are plotted in Appendix A (changed residence since birth) and Figure 4 (same residence since birth). Among the harmful herbicides in the latter stratum, dacthal was the most important with a posterior mean weight of 0.646. For insecticides, weights were evenly distributed across chemicals, as most chemicals had posterior mean weights between 0.035 and 0.060 and cis-Permethrin had the largest posterior mean weight of 0.065.

## 4. Discussion

In this paper, we proposed the Bayesian group index model for chemical mixture analysis for the realistic situation of multiple groups of exposures each with a potentially different magnitude and direction of association with the health outcome. We conducted a simulation study to evaluate the relative performance of the Bayesian group index model and the frequentist approach of GWQS regression and found that the two methods performed similarly for larger studies (n = 1000), but that the Bayesian group index model performed better for smaller studies (n = 500) with smaller strengths of association (OR < 2.0). The Bayesian group index model had more power to find significant exposure effects in smaller studies (with power differences of 0.2 or more). In addition, the Bayesian group index model was more sensitive and more specific than GWQS, particularly for studies with small sample sizes. While the Bayesian approach was more powerful, it also had larger positive bias in effect estimates in the larger studies. Based on the sum of the findings, we recommend use of the Bayesian index model over the GWQS model, particularly for small studies.

For the implementation of the Bayesian group index model in this paper, we used our BayesGWQS R package [24] on The Comprehensive R Archive Network (CRAN). In addition to the BayesGWQS package, our implementation of the GWQS model is also available as an R package entitled GroupWQS [18] along with a vignette on CRAN. From the user’s perspective the packages are similar, utilizing the same workflow and providing tools to organize and then analyze data as well as visualize results. However, the estimation is very different in the two packages. The GroupWQS package first splits the data into training and validation sets, next estimates the index weights of the GWQS model with bootstrap samples of the training set, and then estimates the other model parameters using the validation set. Parameter estimation is done through nonlinear optimization available in the solnp function of the Rsolnp R package. BayesGWQS estimates model parameters by implementing MCMC available in Just Another Gibbs Sampler (JAGS) using all the data. The two packages each offer distinct advantages to researchers depending on the context of their work. GroupWQS tends to have faster runtime, but uses a two-step estimation process. BayesGWQS has a longer runtime, but allows researchers working with smaller sample sizes to maximize power by avoiding data splitting. Currently, both packages require the user to specify the groups of exposures, which could be done based on chemical family, empirical correlations, or another approach to group similar exposures. 

The Bayesian framework for the group index model also allows for more straightforward extension to more complex models that include individual and spatial random effects [20,21,22]. We have previously used both exchangeable and spatially correlated random effects in Bayesian single index models. In addition, imputation of chemical concentrations below the limit of detection can also be accounted for within the Bayesian index model approach. We are currently working on approaches for imputing missing chemical concentrations within the Bayesian group index model.

When applying the Bayesian group index model to observational data from the CCLS, we found a negative and significant association between insecticides (OR = 0.50) and leukemia, with carbaryl (weight = 0.14) being the most important chemical. This finding is consistent with our previous analysis using the frequentist GWQS approach, where we found an OR = 0.43 for insecticides and weight = 0.21 for carbaryl [19]. This finding is similar to results from individual insecticide logistic regression models; however, in both the individual insecticide model and the model adjusted for multiple insecticides, the inverse association with carbaryl was not statistically significant [27]. Additionally, similar were positive yet not significant associations for PCBs (OR = 1.15) and PAHs (OR = 1.16), where we previously found OR = 1.29 for PCBs and OR = 1.31 for PAHs. In the current paper, we conducted a stratified analysis based on the duration of the child’s residence in the home from which the dust sample was collected. We found stronger exposure effects for the children who had lived their entire lives in the home where dust samples were taken. In this set of residentially stable children, there was a strong and significant effect for herbicides (OR = 2.22), with dacthal being the most important exposure in this chemical group (weight = 0.65). This adds to our previous findings about the contribution of dacthal exposure (weight = 0.31 in the full study population) among the herbicides to increased risk (OR = 1.79) of childhood leukemia [19]. However, we did not previously evaluate risk among this group due to the decreased power of GWQS for stratified analyses. In addition to our previous mixture analysis finding, there was significantly elevated risk of acute lymphocytic leukemia (ALL) associated with the presence of dacthal in house dust (detected vs. not detected OR=1.52, 95% CI:1.03, 2.23) in a previous analysis of herbicide exposures in the CCLS [28]. Logistic regression analyses using individual chemicals yielded a positive yet not significant association between dacthal concentration quantiles and ALL risk [28]. While our results suggest some significant associations with environmental chemicals and childhood leukemia, more studies are needed to determine if these findings generalize to other geographic areas. In addition, while we adjusted for several potential risk factors and confounders, residual confounding cannot be ruled out in our analysis.

## 5. Conclusions

In conclusion, our approach of the Bayesian group index model has the potential to make a substantial contribution to the field of environmental epidemiology, particularly for chemical mixture analysis. The method allows for multiple groups of environmental chemical exposures each with a potentially different magnitude and direction of association with the health outcome, and allows for a richer assessment of environmental exposures. Simulation study evaluation shows that it compares favorably with other methods for mixture analysis, especially GWQS regression, and is easily extended to include more complexity in the model. While we applied the method in an environmental chemical risk analysis of childhood leukemia considering different classes of chemicals, it should be applicable to many other diseases with suspected environmental causes. Hopefully, this method will enable investigators to uncover multiple environmental determinants of disease in future studies.

## Figures and Tables

**Figure 1 ijerph-18-03486-f001:**
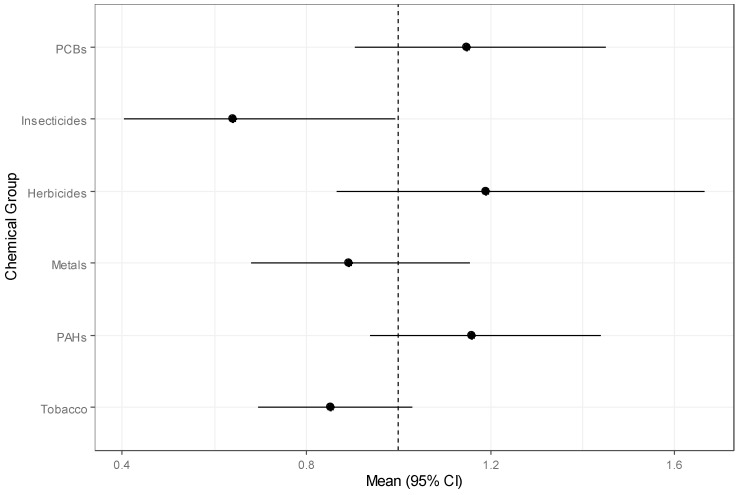
Forest plot of odds ratios and 95% credible intervals for chemical groups for childhood leukemia from the Bayesian group index model with a line at the null value of 1.0.

**Figure 2 ijerph-18-03486-f002:**
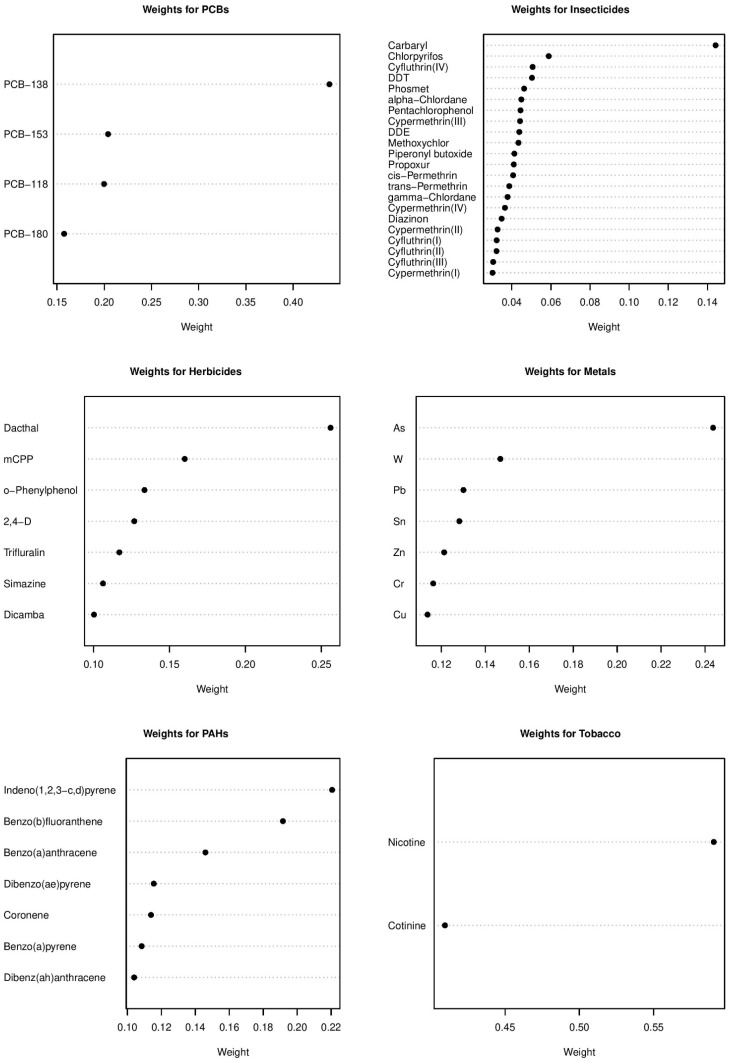
Weights for chemicals in each of the chemical groups from the Bayesian group index model for childhood leukemia in the CCLS.

**Figure 3 ijerph-18-03486-f003:**
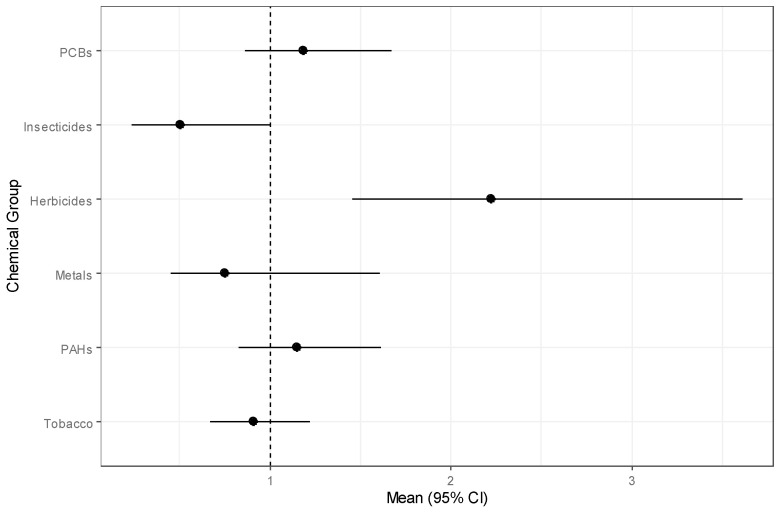
Forest plot of odds ratios and 95% credible intervals for chemical groups for childhood leukemia in children who lived at the same residence since birth with a line at the null value of 1.0.

**Figure 4 ijerph-18-03486-f004:**
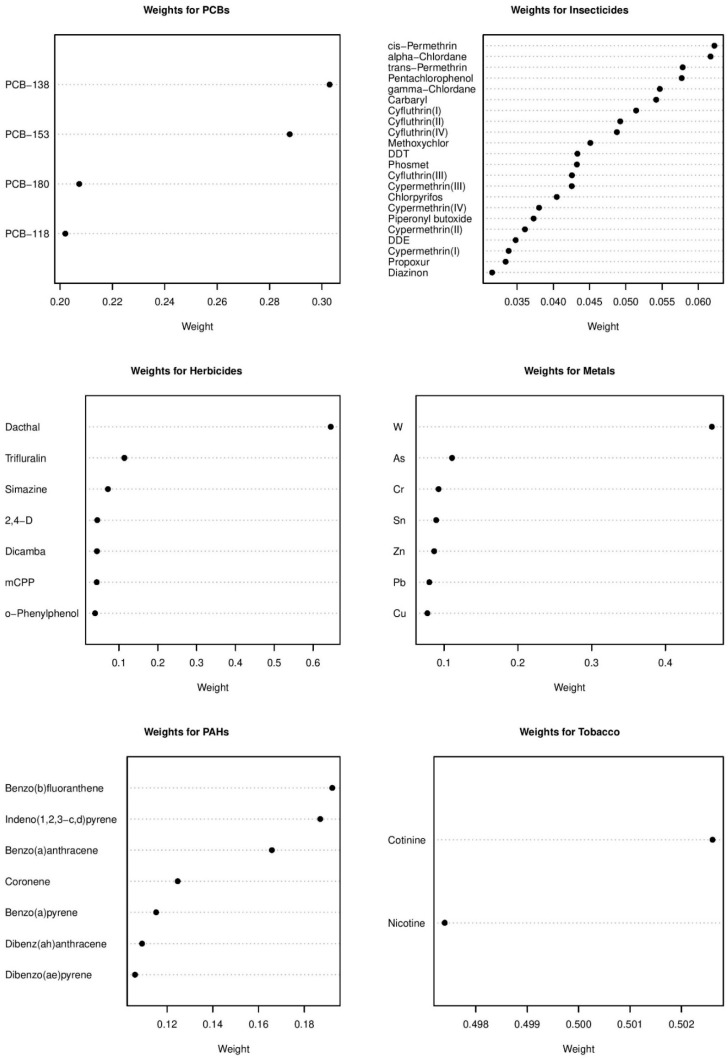
Estimated chemical weights for chemical groups from the Bayesian group index model for childhood leukemia in the CCLS in children with same residence since birth.

**Table 1 ijerph-18-03486-t001:** Definition of the terms used in the simulation study exposure scenarios.

Terms	Levels	Definitions
Exposure Scenario Set	A	9 chemicals; 2 groups (5, 4); 2 important in each group; N = 1000
	B	14 chemicals; 3 groups (5, 4, 5); 1 important in each group; N = 1000
	C	14 chemicals; 3 groups (5, 4, 5); (3, 2, 3) important in each group; N = 1000
	D	14 chemicals; 3 groups (5, 4, 5); (3, 2, 3) important per group; N = 500
Strength of Association	Level 1	OR = 1.00 for all groups (Null effect scenario)
	Level 2	OR = (0.67, 1.50) for A; OR = (0.67, 1.50, 1.50) for B and C; OR = (0.75, 1.25, 1.25) for D
	Level 3	OR = (0.67, 1.50) for A; OR = (0.50, 2.00, 2.00) for B and C; OR = (0.67, 1.50, 1.50) for D
	Level 4	OR = (0.40, 2.50) for A; OR = (0.40, 2.50, 2.50) for B and C; OR = (0.57, 1.75, 1.75) for D
	Level 5	OR = (0.67, 1.50) for A; OR = (0.33, 3.00, 3.00) for B and C; OR = (0.50, 2.00, 2.00) for D
Chemical Correlation Structure	Weak	0.5 within group, 0.1 across group
	Moderate	0.7 within group, 0.3 across group
	Strong	0.9 within group, 0.5 across group

**Table 2 ijerph-18-03486-t002:** Estimated odds ratio (OR) and power values for the Bayesian group index model and group weighted quantile sum (GWQS) regression for Scenario D.

Parameter	Bayesian Group Index		GWQS	
Weak Correlation	Estimated OR	Power	Estimated OR	Power
exp(β_1_) = 1.00	1.004	0.06	1.0306	0.1
exp(β_2_) = 1.00	1.0027	0.07	1.0198	0.06
exp(β_3_) = 1.00	0.9999	0.04	1.0313	0.06
exp(β_1_) = 0.80	0.8364	0.27	0.868	0.13
exp(β_2_) = 1.25	1.2438	0.4	1.2535	0.26
exp(β_3_) = 1.25	1.2506	0.38	1.2304	0.21
exp(β_1_) = 0.67	0.6971	0.77	0.7123	0.53
exp(β_2_) = 1.50	1.5288	0.88	1.5117	0.74
exp(β_3_) = 1.50	1.4756	0.81	1.4867	0.57
exp(β_1_) = 0.57	0.5842	0.98	0.6181	0.8
exp(β_2_) = 1.75	1.8097	1	1.7561	0.91
exp(β_3_) = 1.75	1.6757	0.98	1.6248	0.78
exp(β_1_) = 0.50	0.5096	1	0.5383	0.93
exp(β_2_) = 2.00	2.0995	1	2.0609	0.98
exp(β_3_) = 2.00	1.9448	1	1.9232	0.95
Moderate Correlation				
exp(β_1_) = 1.00	1.001	0.03	1.014	0.04
exp(β_2_) = 1.00	1.0059	0.06	1.0206	0.05
exp(β_3_) = 1.00	1.0024	0.1	1.0079	0.07
exp(β_1_) = 0.80	0.8236	0.44	0.8209	0.29
exp(β_2_) = 1.25	1.2556	0.53	1.2713	0.35
exp(β_3_) = 1.25	1.2339	0.42	1.2313	0.25
exp(β_1_) = 0.67	0.6873	0.91	0.7258	0.62
exp(β_2_) = 1.50	1.5014	0.94	1.4503	0.67
exp(β_3_) = 1.50	1.4834	0.95	1.4836	0.69
exp(β_1_) = 0.57	0.5748	1	0.6094	0.88
exp(β_2_) = 1.75	1.7783	1	1.7019	0.93
exp(β_3_) = 1.75	1.7679	1	1.7434	0.94
exp(β_1_) = 0.50	0.5056	1	0.5369	0.97
exp(β_2_) = 2.00	2.0673	1	2.0541	1
exp(β_3_) = 2.00	2.0072	1	2.0196	1
Strong Correlation				
exp(β_1_) = 1.00	1.0137	0.03	1.0198	0.06
exp(β_2_) = 1.00	0.9879	0.08	1.0023	0.07
exp(β_3_) = 1.00	1.0023	0.03	0.9988	0.06
exp(β_1_) = 0.80	0.8175	0.48	0.8252	0.26
exp(β_2_) = 1.25	1.249	0.52	1.2629	0.35
exp(β_3_) = 1.25	1.2494	0.56	1.2621	0.3
exp(β_1_) = 0.67	0.6705	0.94	0.6847	0.7
exp(β_2_) = 1.50	1.4993	0.96	1.4783	0.71
exp(β_3_) = 1.50	1.5217	0.99	1.5407	0.82
exp(β_1_) = 0.57	0.5797	0.99	0.6078	0.88
exp(β_2_) = 1.75	1.7978	1	1.7951	1
exp(β_3_) = 1.75	1.7145	1	1.6813	0.93
exp(β_1_) = 0.50	0.4987	1	0.5248	0.95
exp(β_2_) = 2.00	2.0279	1	1.9909	0.98
exp(β_3_) = 2.00	2.0217	1	2.0027	1

**Table 3 ijerph-18-03486-t003:** MSE and bias for effect estimates from the Bayesian group index model and group weighted quantile sum (GWQS) regression for Scenario D.

Parameter	Bayesian Group Index		GWQS	
Weak Correlation	MSE	Bias	MSE	Bias
exp(β_1_) = 1.00	0.0148	−0.0034	0.0411	0.0088
exp(β_2_) = 1.00	0.0146	−0.0045	0.0312	0.0041
exp(β_3_) = 1.00	0.0148	−0.0076	0.0302	0.0155
exp(β_1_) = 0.80	0.0146	0.0379	0.0354	0.0663
exp(β_2_) = 1.25	0.0148	−0.0123	0.031	−0.0127
exp(β_3_) = 1.25	0.0166	−0.0080	0.0349	−0.0327
exp(β_1_) = 0.67	0.0231	0.0339	0.0348	0.0503
exp(β_2_) = 1.50	0.0145	0.0119	0.0256	−0.0049
exp(β_3_) = 1.50	0.019	−0.0258	0.0328	−0.0257
exp(β_1_) = 0.57	0.0209	0.0119	0.0421	0.06
exp(β_2_) = 1.75	0.0142	0.0267	0.029	−0.0113
exp(β_3_) = 1.75	0.0194	−0.0518	0.0459	−0.0828
exp(β_1_) = 0.50	0.0203	0.009	0.041	0.0546
exp(β_2_) = 2.00	0.0168	0.041	0.028	0.0162
exp(β_3_) = 2.00	0.0213	−0.03824	0.0429	−0.0598
Moderate Correlation	MSE	Bias	MSE	Bias
exp(β_1_) = 1.00	0.0086	−0.0033	0.0238	0.0015
exp(β_2_) = 1.00	0.0118	0	0.0219	0.0097
exp(β_3_) = 1.00	0.0111	−0.0031	0.0254	−0.0049
exp(β_1_) = 0.80	0.0122	0.0234	0.0217	0.0151
exp(β_2_) = 1.25	0.012	−0.0016	0.0195	0.0073
exp(β_3_) = 1.25	0.0107	−0.0183	0.0209	−0.0252
exp(β_1_) = 0.67	0.0134	0.0241	0.0332	0.0706
exp(β_2_) = 1.50	0.0144	−0.0063	0.0244	−0.0449
exp(β_3_) = 1.50	0.0103	−0.0161	0.0207	−0.0211
exp(β_1_) = 0.57	0.0161	−0.0020	0.0301	0.0501
exp(β_2_) = 1.75	0.0132	0.0095	0.0228	−0.0384
exp(β_3_) = 1.75	0.0151	0.0026	0.0244	−0.0158
exp(β_1_) = 0.50	0.0192	0.0016	0.0402	0.0525
exp(β_2_) = 2.00	0.0166	0.025	0.0288	0.0122
exp(β_3_) = 2.00	0.0174	−0.0052	0.0372	−0.0091
Strong Correlation	MSE	Bias	MSE	Bias
exp(β_1_) = 1.00	0.0089	0.0091	0.0228	0.0084
exp(β_2_) = 1.00	0.0133	−0.0187	0.0268	−0.0112
exp(β_3_) = 1.00	0.009	−0.0022	0.0258	−0.0142
exp(β_1_) = 0.80	0.0118	0.016	0.0253	0.0188
exp(β_2_) = 1.25	0.0114	−0.0065	0.0252	−0.0023
exp(β_3_) = 1.25	0.0102	−0.0055	0.0228	−0.0019
exp(β_1_) = 0.67	0.0131	−0.0007	0.0241	0.0148
exp(β_2_) = 1.50	0.0105	−0.0058	0.023	−0.0258
exp(β_3_) = 1.50	0.0092	0.0098	0.0198	0.0168
exp(β_1_) = 0.57	0.0123	0.0083	0.0301	0.0477
exp(β_2_) = 1.75	0.0126	0.0208	0.0245	0.013
exp(β_3_)= 1.75	0.013	−0.0266	0.0267	−0.0522
exp(β_1_) = 0.50	0.0151	−0.0102	0.0327	0.0323
exp(β_2_) = 2.00	0.0132	0.0073	0.032	−0.0204
exp(β_3_) = 2.00	0.0141	0.0036	0.0258	0.0118

**Table 4 ijerph-18-03486-t004:** Sensitivity and specificity for the Bayesian group index model and group weighted quantile sum (GWQS) regression for simulation scenario D.

		Bayesian Group Index		GWQS	
Effect Size	Correlation	Sensitivity	Specificity	Sensitivity	Specificity
OR = 1.00	Weak	0.381	0.62	0.419	0.618
	Moderate	0.423	0.627	0.4	0.637
	Strong	0.425	0.542	0.38	0.605
OR = 1.50	Weak	0.523	0.793	0.493	0.697
	Moderate	0.504	0.727	0.43	0.695
	Strong	0.496	0.677	0.429	0.693
OR = 2.00	Weak	0.625	0.903	0.618	0.827
	Moderate	0.581	0.835	0.521	0.76
	Strong	0.525	0.753	0.47	0.718
OR = 2.50	Weak	0.685	0.945	0.698	0.873
	Moderate	0.606	0.89	0.574	0.775
	Strong	0.543	0.813	0.523	0.753
OR = 3.00	Weak	0.729	0.963	0.739	0.907
	Moderate	0.673	0.933	0.638	0.842
	Strong	0.583	0.853	0.534	0.757

**Table 5 ijerph-18-03486-t005:** Characteristics of childhood leukemia cases (n = 268) and controls (n = 296) with measurements of chemicals in house dust in the CCLS.

Variable	Controls	Cases
Child’s age, Mean (SD)	3.84 (1.90)	3.77 (1.81)
Female, N (%)	110 (41.0)	121 (40.9)
Child’s Ethnicity, N (%)	130 (43.9)	119 (44.4)
White Non-Hispanic		
Hispanic	101 (34.1)	87 (32.4)
Other Non-Hispanic	65 (22.0)	62 (23.1)
Household Income, N (%)	6 (2.0)	37 (13.8)
Less than USD 15,000		
USD 15,000–29,999	37 (12.5)	27 (10.1)
USD 30,000–44,999	36 (12.2)	44 (16.4)
USD 45,000–59,999	29 (9.8)	33 (12.3)
USD 60,000–74,999	29 (9.8)	20 (7.5)
USD 75,000 or more	159 (53.7)	107 (39.9)
Mother’s education, N (%)	14 (4.7)	16 (6.0)
Less than high school		
High school	60 (20.3)	68 (25.4)
Some college	87 (29.4)	79 (29.5)
Bachelor’s or higher	135 (45.6)	105 (39.2)
Mother’s age, mean (SD)	30.42 (6.30)	30.89 (5.80)
Lived at residence since birth, N (%)	159 (53.7)	120 (44.8)

**Table 6 ijerph-18-03486-t006:** Bayesian group index model odds ratios and 95% credible intervals for chemical groups and demographic variables for childhood leukemia in the CCLS. Bold indicates significant effects according to 95% credible intervals.

Variable	Odds Ratio	2.5% CI	97.5% CI
PCBs	1.15	0.91	1.45
Insecticides	**0.64**	**0.40**	**0.99**
Herbicides	1.19	0.87	1.67
Metals	0.89	0.68	1.15
PAHs	1.16	0.94	1.44
Tobacco	0.85	0.69	1.03
Child’s age	1.01	0.92	1.11
Female	1.00	0.71	1.41
Child’s Ethnicity			
Hispanic vs. White Non-Hispanic	1.22	0.79	1.96
Other Non-Hispanic vs. White Non-Hispanic	1.36	0.88	2.18
Household Income			
USD 15,000–29,999 vs. Less than USD 15,000	0.93	0.42	1.94
USD 30,000–44,999 vs. Less than USD 15,000	0.77	0.35	1.56
USD 45,000–59,999 vs. Less than USD 15,000	0.71	0.30	1.51
USD 60,000–74,999 vs. Less than USD 15,000	0.42	0.17	1.02
USD 75,000 or more vs. Less than USD 15,000	**0.36**	**0.16**	**0.77**
Mother’s education			
High school vs. Less than high school	1.23	0.61	2.73
Some college vs. Less than high school	1.20	0.58	2.73
Bachelor’s or higher vs. Less than high school	1.21	0.57	2.87
Mother’s age	1.02	0.98	1.05
Lived at residence since birth	0.69	0.47	1.01

**Table 7 ijerph-18-03486-t007:** Odds ratios and 95% credible intervals for chemical groups and demographic variables from the Bayesian group index model for subjects with same residence since birth. Bold indicates significant effects according to 95% credible intervals.

Variable	Odds Ratio	2.5% CI	97.5% CI
PCBs	1.19	0.86	1.67
Insecticides	**0.50**	**0.23**	**0.99**
Herbicides	**2.22**	**1.45**	**3.61**
Metals	0.75	0.45	1.61
PAHs	1.15	0.83	1.61
Tobacco	0.91	0.67	1.22
Child’s age	0.87	0.74	1.02
Female	0.99	0.57	1.71
Child’s Ethnicity			
Hispanic vs. White Non-Hispanic	1.27	0.63	2.69
Other Non-Hispanic vs. White Non-Hispanic	1.62	0.84	3.33
Household Income			
USD 15,000–29,999 vs. Less than USD 15,000	1.59	0.48	5.81
USD 30,000–44,999 vs. Less than USD 15,000	0.87	0.26	2.70
USD 45,000–59,999 vs. Less than USD 15,000	0.99	0.28	3.26
USD 60,000–74,999 vs. Less than USD 15,000	0.87	0.23	3.13
USD 75,000 or more vs. Less than USD 15,000	0.37	0.11	1.14
Mother’s education			
High school vs. Less than high school	2.13	0.71	7.96
Some college vs. Less than high school	2.25	0.73	8.79
Bachelor’s or higher vs. Less than high school	1.66	0.51	6.76
Mother’s age	1.04	0.99	1.10

## Data Availability

The CCLS data presented in this study are available on request from the senior author. The data are not publicly available due to privacy restrictions.

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
