# Peer review of "Bayesian Group Index Regression for Modeling Chemical Mixtures and Cancer Risk"

_ijerph, 2021, doi:10.3390/ijerph18073486_

Round 1

Reviewer 1 Report

Initially, I would like to congratulate the authors for addressing a topic of great importance today, the effect of toxic agents such as insecticides in increasing the incidence of chronic diseases, such as cancer. These associations are very difficult to be studied, for several reasons, and the proposed statistical technique allows to deal with one of them, which is the low number of participants.
I congratulate you on the introduction. I think the context, the gap, and the objective were clearly presented.
About the methods, I also found them very well described, but I missed explanations about the implementation of the suggested models, especially about which software was used and if so which packages. In addition, in the supplementary documents, I suggest making available the syntax used, so that it can be easily applied by other researchers. It is also important to mention how the ethical authorization for accessing the data took place.
Regarding the results, I believe that they can be improved, especially the formatting of the graphs, which are presented with the coded names of the variables, such as p.p.ddt, or pcb_138. I suggest editing the graphics with the appropriate labels, as done in figure 1. Also, it would be very important to present the result of the frequent analysis for the data of patients with leukemia, to allow the purchase of the techniques with non-simulated data, even if they are already published in another article. In this sense, that article may be empty without the other data already published.
Regarding the discussion, I believe it is adequate, but I suggest adding the limitations of the study.

Author Response

Reviewer 1

Initially, I would like to congratulate the authors for addressing a topic of great importance today, the effect of toxic agents such as insecticides in increasing the incidence of chronic diseases, such as cancer. These associations are very difficult to be studied, for several reasons, and the proposed statistical technique allows to deal with one of them, which is the low number of participants.

I congratulate you on the introduction. I think the context, the gap, and the objective were clearly presented.

RESPONSE: We thank the reviewer for the positive description of our work.

About the methods, I also found them very well described, but I missed explanations about the implementation of the suggested models, especially about which software was used and if so which packages. In addition, in the supplementary documents, I suggest making available the syntax used, so that it can be easily applied by other researchers. It is also important to mention how the ethical authorization for accessing the data took place.

RESPONSE: We have added details for model fitting using our R packages on lines 208-211 and 276-277. A vignette is freely available on CRAN that demonstrates use of the groupWQS package and the use of the BayesGWQS package follows from this vignette. In addition, the two packages are described in the Discussion (lines 393-409). We have added a statement regarding institutional review board approval on lines 277-280. Note that the line numbers correspond to the tracked change version of the revised manuscript.

Regarding the results, I believe that they can be improved, especially the formatting of the graphs, which are presented with the coded names of the variables, such as p.p.ddt, or pcb_138. I suggest editing the graphics with the appropriate labels, as done in figure 1. Also, it would be very important to present the result of the frequent analysis for the data of patients with leukemia, to allow the purchase of the techniques with non-simulated data, even if they are already published in another article. In this sense, that article may be empty without the other data already published.

RESPONSE: Figure 1 labels are for the chemical class (e.g. PCBs, etc.). Figure 2 labels are the chemical names (e.g., pcb_138, pcb_180). To aid in readability, we have changed the chemical names in supplemental material Table S2 and used the new names in Figures 2 and 4.

Regarding the previous GWQS analysis of the CCLS, the full results are published in Wheeler et al. 2021 (full reference below). We have summarized our new findings and compared them to previous ones in the Discussion, including those from the GWQS analysis (lines 439-465). We did not conduct a stratified analysis using GWQS due to small numbers and the requirement of data splitting with GWQS, and therefore these Bayesian group index model results for the stratified analysis cannot be equally compared with the previous GWQS results. We have noted the addition of the stratified analysis using the Bayesian group index in the section of the text.

Wheeler DC, Rustom S, Carli M, Whitehead T, Ward MH, Metayer C. Assessment of grouped weighted quantile sum regression for modeling chemical mixtures and cancer risk. International Journal of Environmental Research and Public Health 2021; 18(2):504.

Regarding the discussion, I believe it is adequate, but I suggest adding the limitations of the study.

RESPONSE: We have added two limitations to our study in the Discussion (lines 466-470).

Reviewer 2 Report

It was my pleasure to review the manuscript entitled ‘Bayesian Group Index Regression for Modeling Chemical Mixtures and Cancer Risk’, which presents a new approach, involving Bayesian Group Index regression, to examine the association between chemical mixture exposure and its health effects. The manuscript is well organised, and the topic is timely and very important for the related research community. I hope the following comments help authors improve the manuscript.

Major points:

  1. Authors may have chosen an old template. For example, the current format uses numbered references. Please make sure to update the manuscript format.
  2. It is suggested to avoid subjective words such as ‘marginally’ and ‘substantially’. Those wordings may well confuse readers.
  3. Descriptive statistics about variables used to build a model in the CCLS study would be helpful. Especially for stratified analysis, the number of subjects in each group should be presented.
  4. Discussion on how the grouping of chemicals should be done in the Bayesian Group Index regression would be helpful. I think it is a priori or arbitrary but if any recommendations would help readers.

Minor points:

  1. Line 110, ‘wj’ instead of ‘wi’?
  2. Line 125, should it be j1 = 1 (or j2 =1, j3 = 1) in the sigma?
  3. Line 283, ‘difference’ instead of ‘different’
  4. Line 348, ‘considered to show’ instead of ‘considered show’
  5. Figures 1 and 3, add explanation about the slant lines
  6. Table S2, add official chemical names. The first column should be retained since it explains axis legend in Figures 2 and 4.

Author Response

Reviewer 2

It was my pleasure to review the manuscript entitled ‘Bayesian Group Index Regression for Modeling Chemical Mixtures and Cancer Risk’, which presents a new approach, involving Bayesian Group Index regression, to examine the association between chemical mixture exposure and its health effects. The manuscript is well organised, and the topic is timely and very important for the related research community. I hope the following comments help authors improve the manuscript.

RESPONSE: We thank the reviewer for the positive description of our work.

Major points:

Authors may have chosen an old template. For example, the current format uses numbered references. Please make sure to update the manuscript format.

RESPONSE: We have updated to the current template and formatted our references and citations for this template.

It is suggested to avoid subjective words such as ‘marginally’ and ‘substantially’. Those wordings may well confuse readers.

RESPONSE: We have removed the words “marginally” and “substantially” from the text.

Descriptive statistics about variables used to build a model in the CCLS study would be helpful. Especially for stratified analysis, the number of subjects in each group should be presented.

RESPONSE: We have added a new table (Table 5) that provides a descriptive summary of the variables for cases and controls in the CCLS. We also added the counts of subjects for the stratified analysis (lines 268-269).

Discussion on how the grouping of chemicals should be done in the Bayesian Group Index regression would be helpful. I think it is a priori or arbitrary but if any recommendations would help readers.

RESPONSE: We have described the a priori grouping of chemicals based on their use (e.g. insecticides, herbicides) or structural similarity (PCBs, PAH) for the CCLS in lines 255-259. We also added a statement on lines 428-430 regarding grouping the exposures.

Minor points:

Line 110, ‘wj’ instead of ‘wi’?

RESPONSE: We have corrected this text.

Line 125, should it be j1 = 1 (or j2 =1, j3 = 1) in the sigma?

RESPONSE: The notation in the text is correct. The first subscript is for the chemical and the second is for the group.

Line 283, ‘difference’ instead of ‘different’

RESPONSE: “different” is correct.

Line 348, ‘considered to show’ instead of ‘considered show’

RESPONSE: We reworded this sentence to read more clearly.

Figures 1 and 3, add explanation about the slant lines

RESPONSE: We have added text for the reference line at the null value of 1.0.

Table S2, add official chemical names. The first column should be retained since it explains axis legend in Figures 2 and 4.

RESPONSE: We have changed the chemical names in this table and used them in Figures 2 and 4.

Round 2

Reviewer 1 Report

Dear,

Thank you for your careful review of the manuscript and congratulations on the important contribution to the field.

Best regards

This manuscript is a resubmission of an earlier submission. The following is a list of the peer review reports and author responses from that submission.